# Estimating the Risk of Attention Deficit Hyperactivity Disorder (ADHD) in Parents of Children with ADHD and the Association with Their Children’s Disease Severity and Adherence to Medication

**DOI:** 10.3390/children10091440

**Published:** 2023-08-24

**Authors:** Jananheendaran Ragadran, Norazlin Kamal Nor, Juriza Ismail, Jun Jean Ong, Charlotte Sundaraj

**Affiliations:** 1Department of Paediatrics, Universiti Kebangsaan Malaysia Medical Faculty, Kuala Lumpur 56000, Malaysia; 2Department of Paediatrics, School of Medicine, International Medical University, Kuala Lumpur 57000, Malaysia; 3Department of Paediatrics, Hospital Putrajaya, Putrajaya 62502, Malaysia

**Keywords:** ADHD, children, parental ADHD, severity, medication adherence, CAARS

## Abstract

Background: Attention deficit hyperactivity disorder (ADHD) is characterised by inattentiveness, hyperactivity, and impulsivity. Up to half of the affected children have a parent with ADHD. In this study, the risk of ADHD among parents of ADHD children was estimated. The associations between parental ADHD and child ADHD severity and medication adherence were determined. Methodology: Parents of children and adolescents diagnosed with ADHD attending the University Kebangsaan Malaysia Medical Centre (UKMMC) were recruited between June to August 2022 and the administered Conners’ Adult ADHD Rating Scale (CAARS) self-report short form, Vanderbilt ADHD Parent Rating Scale (performance section), and Medication Adherence Report Scale (MARS). Results: Forty-five children with ADHD were recruited and 15 out of 45 (33%) parents were detected to have ADHD. ADHD severity was worse in children with ADHD parents for total severity (mean of 34.67 vs. 29.13, *p* = 0.047) and difficult behaviours at home (mean of 7.87 vs. 6.27, *p* = 0.036). The children’s academic performance and behavioural challenges at home and school were positively correlated with the parental ADHD scores for ‘inattention’ and ‘problems with self-care’ subscales. Conclusions: A total of 33% of ADHD children had parents with ADHD. ADHD children with ADHD parents were more likely to have behavioural problems at home and more severe ADHD. However, no statistical significance was noted with medication adherence.

## 1. Introduction

Attention deficit hyperactivity disorder (ADHD) is a neurodevelopmental disorder characterised by inattentiveness, hyperactivity, and impulsivity with an estimated prevalence of 5.3–7.1% in children and adolescents [1]. A study by Polanczyk and friends assessing worldwide prevalence by means of systemic review was reported to be at 5.29% in 2007 [2,3]. In Malaysia, the prevalence of ADHD has been reported to be 2.75% in boys and 0.6% in girls [4]. ADHD is mainly diagnosed during childhood, commonly below 7 years of age, consisting of either predominantly inattentive or hyperactive-impulsive types or a combined type [4,5].

It has been estimated that approximately 4% of adults in the general population meet the diagnostic criteria for ADHD [6]. In approximately two thirds of children with the disorder, their symptoms persist into adulthood [7,8]. Around 50–90% of children with ADHD are estimated to suffer from functional disability due to their ADHD symptoms in adulthood [9]. In children, the diagnosis is made based on parent and teacher reports using standardised ADHD rating scale questionnaires such as Conners’ ADHD rating scale or the Vanderbilt ADHD rating scale [10]. In adults, the diagnosis is based on the self-rated adult ADHD rating scale questionnaires followed by a clinical interview [10,11].

ADHD has high inheritance, which has been reported to range between 71 and 90% for heritability [12]. Previous studies have reported that up to half of the children with ADHD have a parent with ADHD [2,13]. High risk familial factors directly increase the risk of ADHD diagnosis through the mediating factor of children’s development [14,15].

Many adults with ADHD also remain undiagnosed due to a lack of public and self-awareness, resulting in a delay in seeking medical attention. As a result, they may struggle to cope in managing their home, family, and work life.

Parenting may be affected tremendously in adults with ADHD. It can add to the daily struggle as they grapple with the high needs of parenting as well as attending to their ADHD children’s high-level of needs, even as they try to cope with their own challenges in the work environment and at home. Deficiency in organisational skills, struggling with punctuality, and difficulty in maintaining focus as well keeping emotions in check are well-recognised characteristics of ADHD. These parents may face even more difficulties when living with high expectations in a demanding environment whilst having to deal with their own ADHD characteristics. They may struggle to keep up with their children’s schedules, in maintaining adherence to treatment modalities, and managing their behaviours as parents. Johnstone et al. (2012) [16] found that ADHD symptoms were associated with impairments in effective behavioural control. They reported that higher parental ADHD symptoms correlated with family disorganisation and lax monitoring of their children’s behaviour as well as displaying more inconsistency and over-reactivity during parenting [10,17,18,19,20]. Previous studies have implicated ADHD symptoms in adults with negative outcomes in their children [6,21]. Across studies, parent and child mental illness was found to be associated with additional adversities impacting family functioning and well-being [22,23]. Moreover, parental hostility was noted to contribute to the continuation of ADHD symptoms and lower levels of academic attainment [24]. A study by Agha et al. (2013) [21] found that having a parent with ADHD was associated with a more severe clinical presentation in their ADHD children. This was especially true when the mothers had ADHD, which was associated with an increased severity of overall ADHD in their children. Amongst the reasons postulated for this are shared inherited or environmental risks as well as in gene–environment interplay [21].

A multi-disciplinary outlook is advocated for the management of ADHD, with drug therapy playing a major role [6]. It has been well-documented that parents with ADHD symptoms have difficulty coping with behavioural parent-training therapy at home. Friedman et al. (2020) [25] reported that parental ADHD symptoms were associated with a lack of improvement in their children’s ADHD symptoms as the parents themselves were not able to adhere to behavioural parent-training therapy. Attention, memory, and organisational weaknesses associated with ADHD in these parents were postulated to hinder the consistent application of learned behavioural parent-training skills at home [26]. A recent review of medication adherence studies showed a high level of medication non-adherence among children and adolescents with ADHD [27]. Although there has been limited evidence on the consequences of medication non-adherence and its clinical and functional impact on children with ADHD, poor compliance generally affects the overall behaviour and academic performance [28,29].

## 2. Materials and Methods

### 2.1. Objectives and Hypothesis

The aim of our study was primarily to estimate the risk of ADHD among parents of children and adolescents with ADHD as well as determine the association between the parents’ ADHD diagnosis and the correlation of parental ADHD rating scores with severity in ADHD children. In addition, we sought to determine the association between ADHD diagnosis and the correlation of parental ADHD rating scores with their children’s adherence to ADHD medications.

We hypothesize that parents and children both with ADHD and higher adult ADHD symptom scores are associated with more severe symptoms and poorer compliance to medication in children with ADHD.

### 2.2. Sample Size Calculation

The sample size was calculated based on the formula *n* = [Np(1 − p)]/[(d [30]/Z^2^_1−α/2_ × (N − 1) + p × (1 − p)]. The number of ADHD patients followed-up over a 3-month period under the Child and Adolescent Psychiatry Clinic and Child Development Clinic was 120 children. With a 95% confidence interval level and expected prevalence of parents with ADHD among children with ADHD based on previous studies was 0.4 as well as precision was 5%. Thus, the calculated sample size was 90.

### 2.3. Participants and Procedures

This was a cross-sectional study involving children and adolescents diagnosed with ADHD according to the DSM IV or 5 ADHD criteria, Conners’ ADHD Rating Scales, or Vanderbilt ADHD diagnostic rating scale aged 18 years of age or below, and their parent(s) who were under the follow-up care of the Child and Adolescent Psychiatry Clinic and the Child Development Centre (CDC) at University Kebangsaan Malaysia Medical Centre (UKMMC) [31]. The study participants were recruited via convenience sampling. The data were collected between June and August 2022, and the parent(s) were enrolled after agreeing to the study and written consent was obtained. They were approached for study recruitment while waiting to be seen at clinic appointments and were given the parental questionnaires to be completed. During the clinic session, the attending clinicians were requested to answer questionnaires for their input on child severity and medication compliance. Despite attempting to enrol both parents upon initiation of the study, during the clinic sessions whereby the data collection was carried out, only one parent was present throughout. Efforts were made to reach out to the other parent via Google Form questionnaires that were sent via either email or WhatsApp, but no response was obtained throughout the study period. The exclusion criteria were the presence of autism spectrum disorder or intellectual disability among the ADHD children, parents with severe health or mental disabilities, parents who were not the main caregiver of the child, and non-biological parents. This study received approval from the Research and Ethical Committee of the National University of Malaysia (FF-2020-310).

### 2.4. Measures

Questionnaires were distributed to parents and the attending physicians of children and adolescents with ADHD that were enrolled in this study. The questionnaire package included demographic information and validated questionnaires. The demographic data for the parent and child included basic information such as the highest parental educational level, family income status, and any existing medical or mental health conditions in the parent(s). Next, the parents were asked to answer the validated questionnaires including the Conners’ Adult ADHD Rating Scale (CAARS) self-report short form for themselves, and the performance section of the Vanderbilt ADHD Parent Rating Scale and parent-reported Medication Adherence Report Scale for their children. The child’s attending physician were asked to complete the Clinical Global Impression-Severity (CGI-S) and clinician-standardised questions to assess the severity and medication adherence from the physicians’ point of view.

### 2.5. Parent-Report Questionnaires

The Conners’ Adult ADHD Rating Scale (CAARS) self-report short form [32,33] questionnaire is a validated instrument measuring a range of behaviours including inattention/hyperactivity/impulsivity and problems with self-concept. The instrument also includes the three DSM IV ADHD symptom subscales (inattention and hyperactivity/impulsivity). It is widely used to assess and aid in the diagnosis of ADHD in adults, aged 18 years and above. Parents rate how strongly they agree or disagree with each item on a 4-point Likert scale ranging from 0 (not at all, never) to 3 (very much/very frequently). The CAARS is subdivided into five subscales: inattention, hyperactivity, impulsivity, problems with self-concept, and the ADHD index. Scores >65 were significant clinically and for high risk estimation of ADHD, the requirements were for at least two domains to be scored above 65. The CAARS has been found to have a high internal consistency (α = 0.86 to 0.92) and test–retest reliability (r = 0.80 to 0.91) [25].

In the Vanderbilt ADHD Parent Rating Scale (performance section) [34,35], the parents were requested to report the academic and behaviour performance of their children both at home and at school, based on their school-teacher’s feedback and the parent’s experience. The questions were adapted from the performance section of the Vanderbilt ADHD rating scale, which were available for download via public access at https://www.nichq.org/resource/nichq-vanderbilt-assessment-scales (accessed on 1 July 2022). Items were scored on a Likert scale 1–5 (1 = Excellent, 2 = Above average, 3 = Average, 4 = Somewhat a problem, 5 = Problematic) with a total score of 60.

The medication adherence report scale (MARS) [36,37,38] is a questionnaire covering five common patterns of non-adherence behaviour, scored on a five-point Likert scale (1 = always, 2 = often, 3 = sometimes, 4 = rarely, 5 = never). The parent-rated version was used for this study. Total scores ranged from 5 to 25, with higher scores indicating higher levels of adherent behaviour. A score of 23 or more was defined as high adherence. The MARS has been used to assess medication adherence among ADHD adolescents and children in previous studies and was specifically developed by Thompson et al. (1999) [39,40] to assess drug compliance in psychiatric patients [19,41,42,43].

### 2.6. Physician Assessment Reports

The Clinical Global Impression-Severity (CGI-S) [41,43,44] is a measure that is used as a standard in clinical trials to reflect the physician’s global impression of patient severity (CGI-S). The CGI-S is on a 7-point scale and requires the physician to rate the severity of the patient’s illness at the time of assessment, relative to the physician’s past experience with patients who have the same diagnosis. Ratings range from 1 = normal (no symptoms at all) to 7 = extremely ill (drastically interferes with function); a score of ≥4 was taken as an indication of severe disease.

The clinician standardised questions to elicit medication adherence [40,45] form a list of standard questions provided to the physicians during their interview with the parent and child. The questions approach the issue of adherence indirectly at first through inquiries regarding how the child is doing, how the child feels about the medication, and the presence of any difficulty in taking or administering the medication. More direct questions such as specific details regarding the time taken, dosage, the duration the child has been on medication, and any adult supervision of medication intake are further asked during the interview. The physicians’ conclusions of the patient’s overall adherence to medication based on their own clinical judgement were also scored on a 7-point Likert scale (1—Never takes medication and 7—Always takes medication). Details of each point on the scale were elaborated by the investigator to help physicians make an objective judgement. A score of ≤5 was taken to indicate poor adherence.

### 2.7. Analysis

All analyses were performed using Statistical Package for the Social Sciences (SPSS) version 26. Categorical data were presented as frequency with corresponding percentages, and continuous data as the mean with standard deviation. Chi-squared tests and Student *t*-tests were utilised to determine the association between parental ADHD diagnosis and disease compliance to medication in children with ADHD. The Pearson correlation test was used to study the correlation between the parents’ ADHD rating scale scores and their children’s ADHD severity scores and between the parents’ ADHD rating scale scores and their children’s adherence to ADHD medication. Pearson correlation coefficients of 0.5 and above were taken as strong correlation whereas from 0.3 to 0.49 was taken as moderate, below 0.29 was deemed as weak correlation, and a *p* value of <0.05 was deemed statistically significant.

## 3. Results

A total of 54 subjects were approached during the 3-month study period, of whom one refused and a further eight were excluded based on the exclusion criteria (seven non-biological parents, one parent with underlying psychosis). The final total sample size comprised 45 participants.

Table 1 shows the demographic data of children with ADHD and their parents. Out of 45 children with ADHD, 37 (82.2%) were male and eight (17.8%) were female, with a mean age of 11.7 years, standard deviation of 2.88, and age range between 6 and 17 years of age. The majority of patients were of Malay ethnicity (57.8%). In terms of ADHD medication use, 32 (71.1%) patients were on pharmacotherapy for ADHD (medications included methylphenidate, atomoxetine, or risperidone). We found that 13 out of 15 (86.7%) ADHD patients on medication had parents with ADHD while 19 out of 30 (63.3%) on medication had parents without ADHD. Of the parents who were recruited, mothers made up the bulk of the respondents, with 30 (66.7%) respondents being mothers. The mean age of the parents was 45.04 years, with a standard deviation of 7.7 and an age range from 31 years to 65 years of age. Around 66.7% of the respondent parents had achieved a tertiary education as their highest educational attainment, and only one parent was educated to the primary school level. The majority of families recruited were from the lowest income group ‘B40’ (<MYR 4849) household income group, involving a total of 31 families (68.9%). In terms of previous medical or mental health diagnoses, up to 25 parents (55.5%) had underlying medical or mental health conditions. Of these, 10 (22.2%) had previously been diagnosed with ADHD, one had anxiety disorder, and the remaining 14 were diagnosed with chronic non-communicable diseases such as hypertension and diabetes mellitus.

Table 2 shows the total number of parents who had a previous diagnosis of ADHD, and those who were identified with ADHD using CAARS. The CAARS questionnaire revealed that there were seven parents of ADHD children who were at high risk of ADHD themselves based on their results. Of these, two had already been diagnosed with ADHD. Interestingly, the other five had never been previously diagnosed. The parental ADHD prevalence was 33.3% in this study population. The subscales with the highest scores above the significance level (>65) were ‘ADHD index’ (15.6%), followed by ‘impulsivity’ (13.3%) and ‘inattention’ (11.1%). Those who fulfilled the criteria for ADHD on CAARS all had a significant level of scores for the ‘ADHD index’ subscale, which is a measure that shows that someone is unlikely to have ADHD if the scores are <65; this was the subscale with the most consistently abnormal result. This was followed by the ‘inattention’ subscale at 71%. The remaining 43% was for the ‘hyperactivity’ subscale, with the ‘impulsivity’ subscale at 57% and 14% for the ‘problems with self-concept’ subscale. The most frequent combination of two subscales required for ADHD diagnosis using CAARS were ‘ADHD index’ and ‘inattention’ (71%), followed by the ‘ADHD index’ and ‘impulsivity’ subscales (57%).

Table 3 shows the severity of ADHD in children based on the parent-rated performance section of the Vanderbilt Rating Scale and clinician-rated Clinician Global Impression-Severity (CGI-S).

In Table 3a, the ADHD child severity scores between those with ADHD parents and those who did not have ADHD parents showed that those whose parents had ADHD had higher mean scores in all subscales (academic, school behaviour, and home behaviour), which corresponded to greater severity. For the parent-rated Vanderbilt Rating Scale, the subscales with the lowest mean score were home behaviour at 6.80 (SD: 2.27, range: 3–11). Lower scores are associated with better or more positive outcomes. For academic performance, the mean scores were 11.49 (SD: 3.04, range: 5–19). Behavioural issues at school or ‘school behaviour’ had the highest score with a mean of 12.78 (SD: 4.13, range: 4–22). Both showed poorer outcomes compared to behaviours at home. Children whose parents had ADHD showed greater severity in terms of the total severity (mean of 34.67 vs. 29.13, *p* = 0.047) as well as behaviour at home (mean of 7.87 vs. 6.27, *p* = 0.036) to a statistically significant level compared to those whose parents did not have ADHD. For the subscales of ‘academic’ and ‘school behaviour’, children whose parents had ADHD had higher mean scores, reflecting a greater disease severity than those whose parents did not have ADHD, but not to a statistically significant level.

In Table 3b, for the clinician-rated CGI-S, half of the patients (51%) were deemed to be ‘markedly to severely ill’, as rated by their clinician. Comparing the children whose parents had ADHD with the non-ADHD parental group, the source data did not show greater severity (percentage of 46.7% vs. 53.3%, *p* = 0.67) or a statistically significance level compared to those whose parents did not have ADHD based on the clinician’s rating.

Table 4 shows the medication adherence in ADHD children comparing parents with ADHD and without ADHD based on (a) parent-rated Medication Adherence Report Scale (MARS) and (b) clinician-standardised questions to elicit medication adherence. Overall, it shows that the level of adherence was similar between groups, with 53.1% reported to have good adherence and 46.9% reported to have poor adherence. Parents with ADHD reported that their ADHD children were more likely to have poor medication adherence by 53.8% compared to parents who did not have ADHD, who reported poor medication adherence of 31.6%. Although the proportion of poor medication adherence was greater in the group with ADHD parents, it did not reach statistical significance (*p* = 0.207). For the clinician-rated medication adherence, both groups had similarly good adherence reported at 61.5% and 63.2%, and there was no statistically significant difference between the two groups either (*p* = 0.930).

Table 5 demonstrates the Pearson correlation test between the parental CAARS subscale scores with ADHD severity in the child using parent-rated Vanderbilt performance scores and clinician-rated Clinical Global Impression-Severity (CGI-S). The ‘inattention’ subscale of the parents’ self-rated CAARS was found to be moderately correlated to the statistical significance with all performance measures: the children’s academic performance (r = 0.314, *p* = 0.036), school behavioural issues (r = 0.427, *p* = 0.003), home behavioural issues (r = 0.300, *p* = 0.045), and total severity scores (r = 0.449, *p* = 0.002). The ‘problem with the self-concept’ subscale was also moderately correlated to the statistical significance of *p* < 0.05 with the school, home behavioural, and total severity: school behavioural issues (r = 0.344, *p* = 0.021), home behavioural issues (r = 0.364, *p* = 0.014), and total severity scores (r = 0.363, *p* = 0.014). Finally, the ADHD index subscale similarly had moderate correlation with academic performance, school behaviour, and total severity with statistical significance: the children’s academic performance (r = 0.349, *p* = 0.019), school behavioural issues (r = 0.366, *p* = 0.013), and total severity scores (r = 0.423, *p* = 0.004). There was no correlation observed between the CAARS subscales or total score with the clinician rated CGI-S score. 

Table 6 shows the Pearson correlation analysis between the parental CAARS subscale scores with medication adherence as measured using the parent-rated MARS scores and clinician-rated medication adherence, all of which showed no statistical significance with weak correlation. However, parents with ADHD had a higher percentage of children on medication compared to those whose parents did not have ADHD (86.7% vs. 63.3%) and parents who had ADHD also reported poorer adherence to medication in their ADHD children compared to those parents without ADHD (53.8% vs. 31.6%).

## 4. Discussion

The worldwide prevalence of ADHD in the general population has been reported at 2.8% based on a more recent study by Fayyad et al. in 2016 [46]. Recent literature has found that up to half of the parents of children with ADHD have ADHD themselves [9]. Takeda et al. (2010) [47] reported that in 41–55% of families with at least one child with ADHD, at least one parent was affected. In our study, 10 out of 45 (22.2%) parents with ADHD children reported that they themselves had been diagnosed with ADHD. We went on to assess all of the parents for ADHD features using a recognised and validated tool, the CAARS questionnaire, an instrument used by psychiatrists in our clinical setting to help diagnose ADHD in adults. Based on the CAARS questionnaire, an additional five parents of ADHD children were found to be at high risk of ADHD, and in fact fulfilled the criteria for ADHD as per the questionnaire measures. Including the 10 parents that were previously diagnosed with ADHD, the estimated prevalence of ADHD in this study population was noted to be 15 out of 45, or 33.3%. Starck et al. (2016) [10] reported more fathers with ADHD compared to mothers at 51%. Similarly, in our study, we noted that out of the 15 parents noted to be at high risk of ADHD, fathers were 6-fold higher compared to mothers (86.67% vs. 13.33%), despite the higher levels of respondents who were mothers. Although the results from this study showed parental ADHD prevalence to be slightly less than in other studies, the results still showed that a significant proportion of these parents, especially fathers, had ADHD themselves. This demonstrates the importance of being aware of this association and for clinicians to consider actively screening the parents of ADHD children for ADHD in the parents themselves. This is not only important in terms of managing the child with ADHD better, but also for the overall well-being of the affected parent and the promotion of better overall health in the patient’s family by identifying and offering treatment to individuals who may not yet be aware of their diagnosis and may suffer consequences from it. Interestingly, eight out of 10 parents that were previously diagnosed with ADHD did not have high scores on their CAARS questionnaire. Possible explanations for this are that as they have already been diagnosed, they may have undergone appropriate management and have successfully overcome the many difficulties associated with ADHD and thus currently achieved a good level of functioning. This appears to support the importance of screening and managing parental ADHD, as once it is recognised, effective treatment can be implemented. In fact, the finding that eight of the ten parents who were previously diagnosed with ADHD no longer appeared to fulfil the criteria for ADHD based on the CAARS questionnaire, is encouraging as this suggests that with successful intervention, these parents have a good chance of approaching normative function. The high rate of ADHD in parents of children with ADHD makes it important for managing clinicians to be aware of the substantial risk of ADHD in these parents and to strongly consider screening the parents for ADHD during the initial diagnosis of the children and referring those who fail the screening for appropriate diagnostic work-up and management.

Previously published research has demonstrated the impact of parental ADHD on their children’s ADHD symptoms and their progress [9,48]. Dawson et al. (2016) [32] reported that parental ADHD symptoms were associated with diminished improvements in the children’s ADHD symptoms, academic functioning, social skills, and overall impairment as well as reduced effectiveness in parenting skills. In this study, we found that ADHD children who had parents with ADHD had a greater severity of behavioural issues and for overall ADHD severity. This is not unexpected, as parents with ADHD may not be able to implement appropriate behavioural strategies to help support their children’s development and learning, and the environment the children live in may also be chaotic and not conducive to healthy learning. All of this could lead to a worsening of behaviour, especially at home, as this is the environment whereby the parent has the greatest impact and authority, and thus a parent with ADHD may not be able to provide the best home environment to promote positive behaviours. A study by Friedman et al. (2020) [25] reported that parental ADHD was related to poor application and adherence to behavioural parent training, and that this hindered the child’s improvement and the severity of their behaviours. They reported that parents with high CAARS scores, especially in the subscales of ‘inattention’ and ‘problems with self-concept’, had more severe behavioural issues at home and at school. They also found that children with parents having high ADHD index scores had significant difficulty in their academic performances. In our study, we found that higher ADHD scores for the ‘inattention’ subscale in parents were associated with greater ADHD severity in their children both at home and at school for academic issues as well as for overall difficulty. This is consistent with previous findings and is not surprising as parental ADHD symptoms that affect the parents’ attention, organisational skills, and underlying executive functioning could pose a significant barrier for the effective implementation of behavioural techniques for ADHD management as well as compliance to pharmacotherapy. ADHD parents may not be able to give their full focus and aid in the child’s academic learning due to their own lack of organisation, reduced attention span, and poor time management. In addition, poor behavioural inhibitory skills may give rise to ineffective and inconsistent parental response, further compounded by the disruptive home environment created by parental ADHD [9]. We also found that higher ADHD scores for the ‘problems with self-concept’ subscale in parents were associated with greater ADHD severity in their children both at home and at school as well as for overall difficulty, similar to the findings by Friedman et al. (2020) [25]. Individuals who display poor self-concept often experience negative cognitions and lower belief in self (Richman, Hope, and Michalas, 2010 [49]). Recent evidence (Shaffer and Obradovic, 2017 [48]) showed that diminished parental executive functioning predisposed to decreased positive parenting practices such as providing effective instructions, scaffolding tasks, and greater rates of harsh, reactive parenting. A recent meta-analysis by Park et al. (2022) [19] reported that greater parental ADHD symptoms were associated with less positive and more harsh and lax parenting behaviours. Our findings are consistent with previous published research showing that parental ADHD and symptoms of ADHD in parents are associated with increased severity in their children with ADHD. The correlation between the parents’ ADHD scores and the child’s performance may also be due to the inherited and/or shared environmental factor, though the exact nature of the risk was beyond the scope of this study as we did not analyse the genetic or environmental data.

Agha et al., in 2013, found that parental ADHD scores were correlated with clinician-rated severity [21]. In our study, parental ADHD diagnosis and score were not significantly associated with the clinician-rated severity of the child’s ADHD symptoms. This may be due to the limited sample size, the variability of tools used to assess the severity as well as some subjectivity to the clinical assessment and the difficulty in correlating childhood ADHD severity across potentially different clinicians at different follow-up sessions. In addition, in light of the recent struggles with the pandemic, clinicians may have made some exceptions in terms of what is expected to be ‘acceptable’ in light of the prolonged periods of quarantine at home, resulting in a reduced physical contact time and expected sequelae of the pandemic and lockdown measures.

Medication adherence is linked to parental support for the child to be able to have a structured environment and thus able to organise themselves well enough to remember to take their medications regularly [13,50]. A study by Hong et al. (2013) [28] found that family history of ADHD and parental distress was a strong predictor for medication non-adherence [27]. When we compared the medication adherence as reported by both parents and clinicians, higher parental scores for ADHD did not seem to be significantly associated with poorer adherence. In our study, 53.1% of parents and 62.5% of clinicians felt that their medication adherence was good. This was more than half of the study group, but still far from achieving full adherence. The ideal goal was to have close to 100% medication adherence for the better management of ADHD symptoms for all patients. Of interest, in the group with ADHD parents, 13 out of 15 (86.7%) were on medication for ADHD compared to 19 out of 30 (63.3%) children in the group with non-ADHD parents. This suggests that ADHD children whose parents also have ADHD have either more severe symptoms, poorer parenting, greater challenges in implementing successful behavioural strategies, or a combination of these factors. Comparing parents with ADHD and parents without ADHD for medication adherence, 53.8% of parents in the ADHD group deemed that their children were poorly adherent to medication compared to 31.6% from the non-ADHD group. Although this finding did not reach statistical significance, it is still worthwhile noting that the group with ADHD parents reported poor adherence compared to the group whose parents did not have ADHD. In 2013, Hong et al. [28] found that family history of ADHD was a strong predictor for medication non-adherence. We postulate that the limited sample size and different tools used to measure adherence may have impacted our results, although the trend in our population suggests poorer compliance in the group with parental ADHD. Moreover, the Medication Adherence Rating Scale (MARS) is a parent-rated tool and responses may be subjected to reporting bias, as parents may be reluctant to give unfavourable responses to their physician out of fear of prejudice or criticism.

## 5. Limitations and Future Directions

We recognise that our study had its limitations. First, the small sample at only 50% might not be powered to ascertain differences for various aspects of the study, especially associations with small effect sizes. There may have also been reporting bias from the parents as well as clinicians due to the subjective perception of performance. Although CAARS is a diagnostic tool used in our clinical setting, it is often used in conjunction with clinical assessment. In this study, a full diagnosis of the at-risk parents was not performed and only the CAARS score was utilised. Thus, the measured value may not truly reflect the true prevalence of ADHD in this population. Furthermore, parents who scored a high CAARS score would require full formal assessment by a trained clinician in order to have a recognised diagnosis of ADHD. Finally, we did not examine both of the parents’ ADHD rating scores, and thus the results were limited to the one available parent that was recruited into the study. The other parent that was not evaluated may have had significant scores that could have the potential significance on the severity of ADHD and adherence to the behavioural, pharmacotherapy, and other modalities of management in ADHD children.

In the future, to improve this study, a larger sample size from a more representative sample including other centres should be obtained. Both parents should be assessed for ADHD risk and potentially compared. A more comprehensive assessment of the child’s severity with measures from not only parents, but teachers would make the severity assessment of the children more robust.

## 6. Conclusions

The total prevalence of parental ADHD in this study was estimated at 33%. There is a significant risk of ADHD in parents of children with ADHD and it may adversely impact the child’s condition and management. Clinicians should strongly consider screening parents for ADHD when a child is diagnosed with ADHD. ADHD in parents is associated with worse outcomes in children with ADHD at home, at school, and for overall functioning. High scores for inattention, problems with self-concept, and ADHD index in the parents’ ADHD subscales correlate with a greater severity of behaviour problem at both school and home as well as overall difficulties for children with ADHD.

## Figures and Tables

**Table 1 children-10-01440-t001:** Characteristics of children and adolescents with ADHD and their parents.

Characteristics	Total (N = 45)		ADHD (N = 15)		Non-ADHD Parents (N = 30)	
ADHD Child or Adolescent	N	%	N	%	N	%
Age (years)						
Mean	11.67	-	11.67	-	11.67	-
Standard deviation	2.88	-	2.60	-	3.06	-
Range (min–max)	6–17	-	8–15	-	6–17	-
Gender, n (%)						
Male	37	82.2	13	35.1	24	64.9
Female	8	17.8	2	25	6	75
Race, n (%)						
Malay	26	57.8	8	30.8	18	69.2
Non-Malay	19	42.2	7	36.8	12	63.2
ADHD Medication						
On ADHD meds	32	71.1	13	40.6	19	59.4
Not on ADHD meds	13	28.9	2	15.4	11	84.6
Parent						
Age (years)						
Mean	45.04	-	46	-	44.57	-
Standard Deviation	7.70	-	6.92	-	8.13	-
Range	31–65	-	36–58	-	31–65	-
Gender						
Male (Fathers)	15	33.3	6	40	9	60
Female (Mothers)	30	66.7	9	30	21	70
Highest Education Attainment, n (%)						
Primary	1	2.2	0	0	1	100
Secondary	14	31.1	5	35.7	9	64.3
Tertiary	30	66.7	10	33.3	20	66.7
Household Income, n (%)						
<MYR 4849 (B40)	31	68.9	10	32.3	21	67.7
MYR 4850–10,959 (M40)	5	11.1	2	40	3	60
>MYR 10,960 (T20)	9	20	3	33.3	6	66.7
Parental Medical/Mental Health Diagnoses, n (%)						
None	20	44.4				
Present	25	55.6				
ADHD	10	40				
Mental Disorders	1	4				
Others (hypertension, diabetes mellitus, asthma, hepatitis B, rheumatic heart dis.)	14	56				

**Table 2 children-10-01440-t002:** Parents with ADHD based on previous diagnosis and CAARS score.

ADHD in Parents	N	%
Previously diagnosed with ADHD	10	22.2
High-risk of ADHD due to significant CAARS score (≥2 subscales of score ≥66)	7	15.6
Parents previously diagnosed with ADHD and have significant CAARS score (overlap of two categories above)	2	-
Total number of parents with ADHD based on previous diagnosis and by CAARS score	15	33.3

**Table 3 children-10-01440-t003:** Severity of ADHD in children of parents with ADHD and without based on (**a**) parent rated Vanderbilt performance section and (**b**) clinician rated CGI-S.

(a)										
Vanderbilt Performance Scores by Subscales	All (N = 45)			ADHD in Parent (N = 15)			No ADHD in Parent (N = 30)			*p*-Value
	Mean	SD	Range	Mean	SD	Range	Mean	SD	Range	
Academic	11.49	3.04	5–19	12.6	3.18	8–19	10.93	2.86	5–19	0.172
Home Behavioural	6.80	2.27	3–11	7.87	2.10	3–11	6.27	2.20	3–11	0.036 *
School Behavioural	12.78	4.13	4–22	14.20	4.06	6–20	12.07	4.05	4–22	0.194
Total	30.98	7.83	9–48	34.67	7.99	17–48	29.13	7.19	9–41	0.047 *
**(b)**										
**CGI-S**	**All (N = 45)**			**ADHD in Parent (N = 15)**			**No ADHD in Parent (N = 30)**			***p*-Value**
	**N**	**%**		**N**	**%**		**N**	**%**		
Low Severity, n (%)	22	48.9		8	53.3		14	46.7		0.670
High Severity, n (%)	23	51.1		7	46.7		16	53.3		

(a) Student *t*-Test, * Statistical significance with *p* < 0.05. (b) Chi-squared test, * Statistical significance with *p* < 0.05.

**Table 4 children-10-01440-t004:** Medication adherence in ADHD children comparing parents with ADHD and without ADHD based on (a) parent-rated Medication Adherence Report Scale (MARS) and (b) clinician standardised questions to elicit medication adherence.

(a)							
MARS	All (N = 32)		ADHD (N = 13)		Non-ADHD (N = 19)		*p*-Value
	N	%	N	%	N	%	
Good Adherence (Score ≥ 23), n (%)	17	53.1	6	46.2	13	68.4	0.207
Poor Adherence (Score < 23), n (%)	15	46.9	7	53.8	6	31.6	
**(b)**							
Good Adherence (Score ≥ 5), n (%)	20	62.5	8	61.5	12	63.2	0.930
Poor Adherence (Score < 5), n (%)	12	37.5	5	38.5	7	36.8	

*p*-value using the Chi-squared test.

**Table 5 children-10-01440-t005:** Correlation of the parents’ CAARS scores with ADHD severity based on parent-rated Vanderbilt performance section and clinician rated CGI-S.

Pearson Correlation of Parental CAARS Subscales with Child Severity Using Vanderbilt and CGI-S							
			Parent-Rated Vanderbilt Performance Section				Clinician-Rated CGI-S
Parental CAARS Subscales	Inattention	Pearson Correlation	0.314 *	0.427 **	0.300 *	0.449 **	−0.190
		Sig.(2-tailed)	0.036	0.003	0.045	0.002	0.212
	Hyperactivity	Pearson Correlation	0.265	0.166	0.130	0.220	−0.199
		Sig.(2-tailed)	0.079	0.276	0.396	0.146	0.190
	Impulsivity	Pearson Correlation	0.176	0.179	0.208	0.217	−0.031
		Sig.(2-tailed)	0.248	0.240	0.170	0.152	0.840
	Problems with Self-Concept	Pearson Correlation	0.199	0.344 *	0.364 *	0.363 *	−0.009
		Sig.(2-tailed)	0.190	0.021	0.014	0.014	0.951
	ADHD Index	Pearson Correlation	0.349 *	0.366 *	0.289	0.423 **	−0.026
		Sig.(2-tailed)	0.019	0.013	0.054	0.004	0.868

* Significant for *p* < 0.05; ** Significant for *p* < 0.005.

**Table 6 children-10-01440-t006:** Correlation of the parents’ CAARS scores with child medication adherence based on parent-rated Medication Adherence (MARS) and clinician standardised questions to elicit medication adherence.

Pearson Correlation of CAARS with MARS and Clinician-Rated Medication Adherence				
			Parent-Rated Medication Adherence (MARS)	Clinician-Rated Medication Adherence (CSqMA)
CAARS Subscales	Inattention	Pearson Correlation	0.081	0.013
		Sig. (2-tailed)	0.659	0.944
	Hyperactivity	Pearson Correlation	0.153	0.011
		Sig. (2-tailed)	0.403	0.953
	Impulsive	Pearson Correlation	0.270	−0.033
		Sig. (2-tailed)	0.136	0.858
	Problems with Self Concept	Pearson Correlation	0.135	−0.056
		Sig. (2-tailed)	0.460	0.762
	ADHD Index	Pearson Correlation	0.255	−0.129
		Sig. (2-tailed)	0.160	0.481

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
