# Peer review of "Estimating the Risk of Attention Deficit Hyperactivity Disorder (ADHD) in Parents of Children with ADHD and the Association with Their Children’s Disease Severity and Adherence to Medication"

_children, 2023, doi:10.3390/children10091440_

Round 1

Author Response

Thank you very much for your feedback, pertaining the issues raised. we have attached the replies on the feedback on the attached word document . thank you

Reviewer 2 Report

Comments have been added as a word file.

Author Response

Thank you for taking your time to reviewing this article, we appreciate your input to further improve the article. The following are responses to the issues raised. Thank you

Reviewer 3 Report

In this interesting article, the authors present an analysis of potential links between ADHD in children and the presence of ADHD in their parents.  Drawing participants from a clinic that assesses 120 patients over a 3 month period, the authors endeavoured to recruit parent and child, with an intended group of 90 to satisfy power laws of analysis.  Unfortunately, they were only able to recruit 54, with a total number of 45 taking part in the experimental paradigm.   Nevertheless, the study found that there was a higher risk of ADHD in parents of children with ADHD, and that their behaviour at home and in adherence to their medication regime was significantly poorer than the other ADHD group.

This is an interesting article which has something important to contribute, but I would advise the authors to collect some more data using the same tools over the next couple of months, in order to be nearer to the proposed number of participants.  It was interesting to note that the majority of adults with a diagnosis of ADHD were no longer identified as clinically at risk by the adult questionnaires, and also that it was mainly the fathers who showed symptoms of ADHD, not the mothers, although it was typically the mothers who took part in the study.  It would be more useful to demonstrate current functioning in the fathers, given the low incidence of ADHD in girls reported here. Should the reader conclude that the mothers answered the questionnaires on behalf of the fathers? Please clarify.

Finally, the authors have struggled with some of the requirements of writing the article.  The section on the participants should include the 1st half of table 1, (age, gender and race), demographics that are not affected by any experimental manipulations.  The remainder fits well within the results section as currently shown.

In summary a promising article that would benefit from the inclusion of further participants to strengthen the argument, as well as some clarification on the mismatch between mothers and fathers in completing the scales.

Author Response

we appreciate your input and we have addressed the issues raised in the word attachment below. thank you

Round 2

Reviewer 2 Report

Thank you for your valuable answers. In my opinion, the study in its current form is not suitable for publication due to methodological errors and the direct impact of these errors on the study design.

Author Response

Dear reviewer, 

Thank you for your very valuable input. We have attempted to answer the specific questions and issues you mentioned and to amend the manuscript to our best ability based on the 12 points you raised. We made the amendments you recommended and responded to your questions point by point. You mentioned following our revisions that "In my opinion, the study in its current form is not suitable for publication due to methodological errors and the direct impact of these errors on the study design." We would like to humbly request for your assistance to point out and help us understand what the methodological errors were and how these errors impacted the study design? This would improve our scientific vigour and benefit us greatly as researchers. If there is any way we can address these specific concerns, we would be grateful to be given the opportunity to do so. 
